# Improvement in Vascular Endothelial Function following Transcatheter Aortic Valve Implantation

**DOI:** 10.3390/medicina57101008

**Published:** 2021-09-24

**Authors:** Shuhei Tanaka, Teruhiko Imamura, Ryuichi Ushijima, Mitsuo Sobajima, Nobuyuki Fukuda, Hiroshi Ueno, Tadakazu Hirai, Koichiro Kinugawa

**Affiliations:** 1Second Department of Internal Medicine, University of Toyama, Toyama 930-0194, Japan; stanaka@med.u-toyama.ac.jp (S.T.); ryuushi@med.u-toyama.ac.jp (R.U.); soba1126@yahoo.co.jp (M.S.); nfukuda@med.u-toyama.ac.jp (N.F.); hiroshi.ueno.md@gmail.com (H.U.); kinugawa0422@gmail.com (K.K.); 2Fujikoshi Hospital, Toyama 930-0964, Japan; thiraitoyama@yahoo.com

**Keywords:** vascular endothelial function, FMD, aortic stenosis, heart failure

## Abstract

*Background and objectives*: Endothelial dysfunction is associated with exercise intolerance and adverse cardiovascular events. Transcatheter aortic valve implantation (TAVI) is applied to treat elderly patients with severe aortic stenosis, but less is known about the impact of TAVI on endothelial dysfunction, which can be assessed by measuring flow-mediated vasodilation (FMD). In this parameter, a low value indicates impaired endothelial function. *Materials and Methods:* Vascular endothelial function was evaluated by FMD of the brachial artery just before and one week after TAVI. Factors associated with the normalization of FMD and their prognostic impact were investigated. *Results*: Fifty-one patients who underwent TAVI procedure (median 86 years old, 12 men) were included. FMD improved significantly from baseline to one week following TAVI (from 5.3% [3.7%, 6.7%] to 6.3% [4.7%, 8.1%], *p* < 0.001). Among 33 patients with baseline low FMD (≤6.0%), FMD normalized up to >6.0% following TAVI in 15 patients. Baseline higher cardiac index was independently associated with normalization of FMD following TAVI (odds ratio 11.8, 95% confidence interval 1.12–124; *p* < 0.04). *Conclusions*: Endothelial dysfunction improved following TAVI in many patients with severe aortic stenosis. The implication of this finding is the next concern.

## 1. Introduction

Severe aortic stenosis (AS) is associated with limited cardiac output, impaired functional capacity, and poor outcomes [1]. Transcatheter aortic valve implantation (TAVI) has demonstrated greater clinical outcomes than surgical aortic valve replacement (SAVR), and it has become realistic to treat elderly patients with AS thus far [2]. Compared to SAVR, TAVI is less invasive because it does not require thoracotomy and cardiopulmonary bypass and can preserve patients’ daily living activities.

Vascular endothelial dysfunction develops early in the pre-stage of arteriosclerosis and its assessment is practical for the early detection of arteriosclerosis. One of the major indices of vascular endothelial function is flow-mediated vasodilatation (FMD). FMD consists of (1) increased blood flow after the release of ischemia due to cuff release, (2) endothelium-derived nitric oxide (NO) release triggered by incremental shear stress, and (3) vasodilation caused by the NO-related relaxation of the vascular smooth muscle [3]. The low value of FMD is associated with aging, obesity, smoking, and hypertension, as well as the progression of cardiovascular diseases [4]. It has been reported that patients with improved FMD after interventions to cardiovascular diseases such as coronary artery disease and hypertension had higher survival rates than those with persistently impaired FMD [5,6].

The progression of AS not only has negative impacts on the left ventricle but also on vascular function, including systemic vascular compliance [7]. Although little has been reported, FMD seems to improve at several years following SAVR or TAVI [8,9]. However, immediate changes in FMD following TAVI and baseline factors associating with post-TAVI improvement in FMD remain unknown.

## 2. Materials and Methods

### 2.1. Patient Selection

Patients who were admitted to our institute to receive TAVI between June 2015 and January 2019 were included in this prospective study. TAVI candidates were assigned to one of three teams randomly during the index hospitalization. Patients assigned to a certain team were finally included in this study and received FMD measurement just before and one week after TAVI.

Patients with symptomatic severe AS who had a high risk of surgical procedure were indicated to receive TAVI determined by the heart valve team. All procedures were performed with a balloon-expandable valve (Sapien XT or Sapien 3; Edwards Lifesciences Inc., Irvine, CA, USA) or self-expandable valve (CoreValve or Evolut R; Medtronic Inc., Minneapolis, MN, USA) via a trans-femoral approach or trans-apical approach under general or local anesthesia. Informed consent was obtained beforehand, and this study was approved by the local institutional review board on 13 August 2018 (IRB 30-415).

### 2.2. Data Collection

Preoperative baseline characteristics including demographics, transthoracic echocardiographic findings, and hemodynamic data were collected. Postoperative data including laboratory and echocardiography data were collected one week following TAVI. All-cause death during the two-year observational period following the index discharger was counted. 

### 2.3. FMD Measurement

Vascular endothelial function was evaluated by FMD of the brachial artery just before and one week after TAVI. Patients rested at least for 10 min in the supine position beforehand in a quiet, light- and temperature-controlled room. Vasodilatation responses were determined by the ultrasound technique using a semi-automatic device (EF18G; UNEX, Nagoya, Japan) by experts who were blinded to the clinical data. Briefly, the diameter of the brachial artery was measured from B-mode ultrasound images using a 10-MHz linear array transducer. Then, a blood pressure cuff on the forearm was inflated to 50 mmHg above the systolic blood pressure for 5 min. The diastolic diameter of the brachial artery was semi-automatically followed using an instrument equipped with software for two minutes. The changes in the diastolic diameter were continuously recorded and the maximum diastolic diameter of the brachial artery was determined. FMD was determined as the maximum change in diameter after cuff release.

### 2.4. Statistical Analyses

Statistical analyses were performed with JMP^®^ Pro 15 (SAS Institute Inc., Cary, NC, USA). Two-sided *p*-values < 0.05 were considered statistically significant. The primary outcome was the change in FMD following TAVI. The secondary outcome was the impact of FMD normalization on two-year survival. 

Continuous variables were expressed as median and interquartile and compared between the two groups using the Mann–Whitney U test. Categorical variables were expressed as numbers and percentages and compared between the two groups using Fischer’s exact test. Wilcoxon signed-rank test was performed to compare the coupled data such as FMD before and after TAVI. Logistic regression analyses were performed to investigate baseline characteristics associating with the normalization of FMD, defined as FMD >6.0%, among those with baseline abnormal FMD. Multivariate analysis was performed using parameters with *p* < 0.20 in the univariate analyses.

## 3. Results

### 3.1. Baseline Characteristics

Among 181 patients, 51 received FMD measurements and were included in this study. The baseline characteristics of all 51 patients are summarized in Table 1. The median age was 86 (82, 89) years and median STS score was 5.1% (3.5%, 7.9%). The median FMD before TAVI was 5.3% (3.2%, 6.7%). The median plasma B-type natriuretic peptide (BNP) level was 297 (114, 620) pg/mL, and all patients had symptomatic severe AS with an aortic valve area <1.0 cm^2^. Left ventricular ejection fraction and cardiac index were preserved as 65% (57%, 72%) and 2.64 (2.28, 2.96) L/min/m^2^ in most of the patients.

Thirty-three patients (65%) had FMD < 6.0% at baseline and were assigned to the low FMD group. The low FMD group had more men and less advanced AS (*p* < 0.05 for both; Table 1). There were no statistically significant differences in other baseline characteristics.

### 3.2. Changes in Clinical Parameters following TAVI

The effective orifice area was significantly improved 1 week after TAVI (from 0.57 cm^2^ to 1.47 cm^2^, *p* < 0.001). Echocardiographic parameters remained unchanged except for aortic valve data. Systolic blood pressure increased, diastolic blood pressure remained unchanged, and pulse pressure increased. Plasma BNP levels decreased significantly (from 368 to 138 pg/mL, *p* < 0.001) (Table 2).

### 3.3. Changes in FMD Immediately after TAVI Procedure

In total, FMD was significantly improved at 1 week following the TAVI procedure (from 5.3% [3.7%, 6.7%] to 6.3% [4.7%, 8.1%], *p* < 0.001; Figure 1).

Among those with low FMD at baseline (*n* = 33/51), 15 patients achieved a normalization of the FMD value (from 5.1% [4.3%, 5.4%] to 7.4% [6.5%, 8.0%], *p* < 0.001), whereas FMD remained at low values in the remaining 18 patients (FMD from 3.2% [2.3%, 5.2%] to 4.4% [2.8%, 5.1%], *p* = 0.87) (Figure 2).

### 3.4. Normalization of FMD after TAVI

Among baseline characteristics, variables with *p* < 0.20 in the univariate analysis including cardiac index, plasma BNP level, peak velocity at aortic valve, and the use of angiotensin converting enzyme inhibitor, were enrolled in the multivariate analysis. Finally, a higher cardiac index at baseline was an independent predictor of FMD normalization (odds ratio 11.8, 95% confidence interval 1.12-124, *p* = 0.040; Table 3).

A cutoff of baseline cardiac index to predict FMD normalization was calculated as 2.94 L/min/m^2^ by the receiver operating characteristics analysis. Patients with a baseline cardiac index above the cutoff experienced a significant improvement in FMD (from 5.0% [3.8%, 5.3%] to 6.3% [6.2%, 6.6%], *p* = 0.002; Figure 3). FMD remained unchanged in those with a baseline cardiac index equal or below the cutoff (from 4.4% [2.4%, 5.3%] to 5.3% [3.6%, 7.4%], *p* = 0.65; Figure 3).

### 3.5. Normalization of FMD and Survival

In-hospital mortality after TAVI was 0%, and there were only three all-cause deaths within two years (5.9%). Two of these deaths were in the baseline low FMD group, one of which was cardiac death. There were no heart failure readmissions. Patients with normalized FMD (*n* = 15) tended to have a higher two-year survival rate compared with those with persistently low FMD (*n* = 18) (100% vs. 89%, *p* = 0.18; Figure 4).

## 4. Discussion

In this study, we investigated the change in FMD following TAVI. FMD increased significantly at one week following TAVI in the overall cohort. Of note, 15 patients achieved a normalization of FMD (>6.0%) among 33 patients with baseline low FMD (<6.0%). A higher cardiac index at baseline was associated with the achievement of normalization in FMD following TAVI. Normalized FMD tended to be associated with higher survival compared with persistently low FMD. 

### 4.1. Endothelial Dysfunction and Severe AS

AS is one of the expressions of systemic atherosclerosis. Atherosclerosis accompanies vascular endothelial cell damage, which is facilitated by the existences of hypertension, hyperglycemia, hyperlipidemia, and physical irritation. In the progression of atherosclerosis, physiologically active substances such as PDGF are released to form initial lesions [10]. Similar findings are observed in the early stage of AS [11]. 

Furthermore, the progression of AS increases wall shear stress due to fast systemic blood flow [12]. Increased shear stress activates endothelium NO synthase and facilitates NO production [13]. As a result, NO expression is continuously elevated in patients with severe AS. In such a situation, further increase in NO activity during the hyperemia would be poor. This is a rationale as to why FMD is low in patients with severe AS.

### 4.2. Change in FMD following TAVI

The improvement of AS reduces wall shear stress at rest. Following TAVI, the changes in wall shear stress following the cuff release become significant. This was the dominant mechanism of improvement in FMD following the intervention in the overall cohort. Both TAVI and SAVR seem to improve vascular endothelial function at least in the chronic phase [8,9]. 

The impact of intervention to AS on FMD levels in the early phase remains controversial. FMD seems to decrease soon after SAVR [14], probably due to the use of cardiopulmonary bypass, which seems to have a negative impact on NO expression and its bioavailability [15]. TAVI, which does not use cardiopulmonary bypass system, might be advantageous in preventing peri-procedural endothelial damage as observed in SAVR. 

Other groups reported the opposite finding to ours, in that wall share stress was impaired soon after TAVI [16]. We found that a baseline preserved cardiac output was the key to achieve an improvement in FMD following TAVI. Not all participants but a specific cohort might enjoy considerable improvement in endothelial function following TAVI. Further studies are warranted to find such responders.

### 4.3. Limitations

First, this is a single-center analysis among a small-sized cohort. Second, there might be intra- and inter-rater variability in the measurements of FMD. In this study, a single experienced expert operator performed all procedures. Third, we investigated the impact of normalization in FMD on survival, but its impact on other clinical outcomes remains the next concern. Fourth, those with improvement in FMD tended to be non-smokers (87% vs. 61%), although this did not reach statistical difference (*p* = 0.10). We cannot completely deny any factors other than cardiac output that might have a considerable impact on the improvement in FMD. Fifth, most patients in the baseline normal FMD were female (17/18). The progression process of endothelial dysfunction might be different between men and women, and further sub-analysis between men and women might bring us further insights.

## 5. Conclusions

Endothelial function, assessed by using FMD, improved immediately after TAVI. Preserved cardiac output might be a key to improve FMD. The clinical implication of improvement in FMD following TAVI remains the next concern.

## Figures and Tables

**Figure 1 medicina-57-01008-f001:**
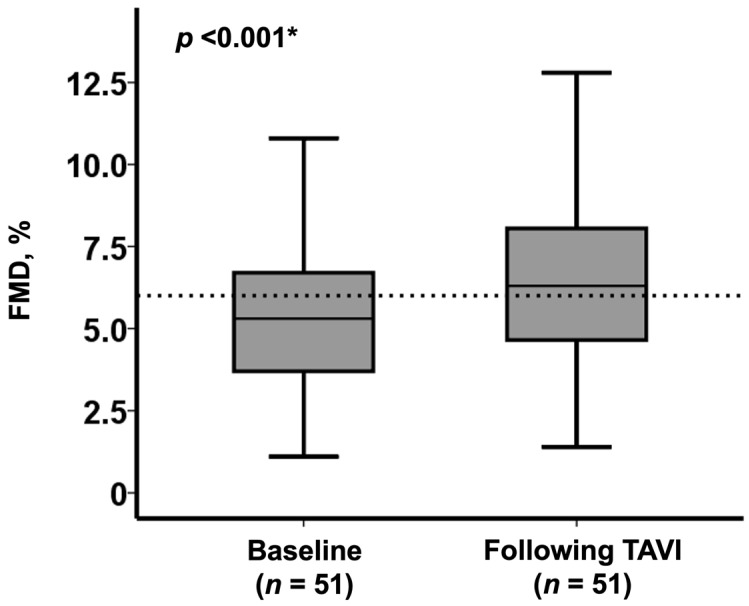
Change in FMD following TAVI in the overall cohort (*n* = 51) * *p* < 0.05 by Wilcoxon signed-rank test.

**Figure 2 medicina-57-01008-f002:**
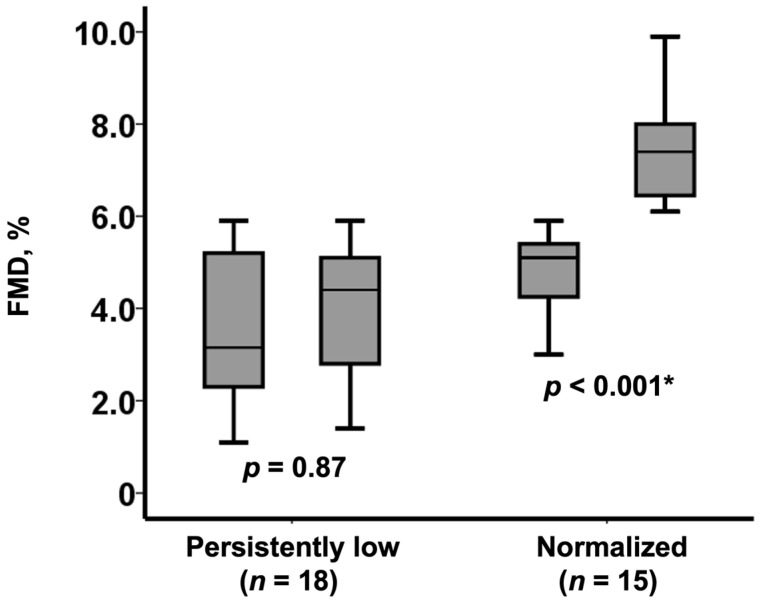
Change in FMD following TAVI among those with baseline low FMD (*n* = 33). FMD remained persistently low in 18 patients whereas normalized in 15 patients. * *p* < 0.0.5 by Wilcoxon signed-rank test.

**Figure 3 medicina-57-01008-f003:**
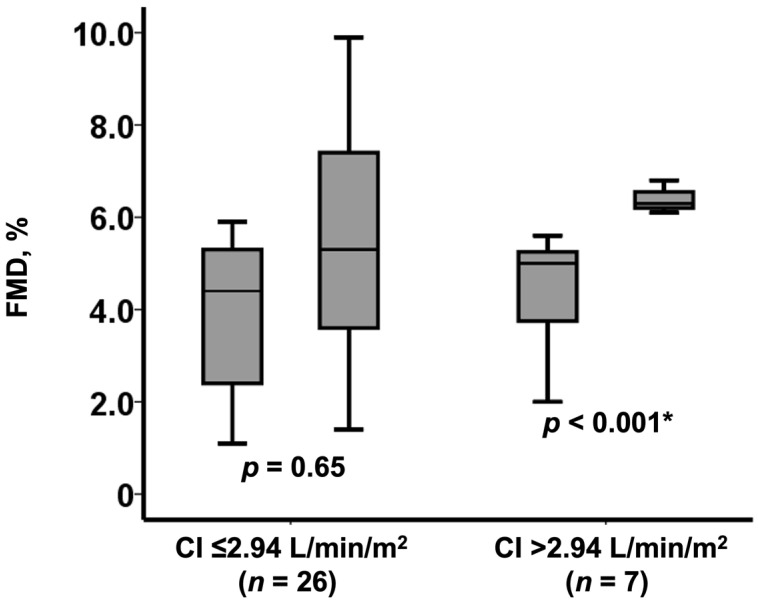
Change in FMD following TAVI among those with baseline low FMD (*n* = 33) stratified by the baseline cardiac index level. FMD remained low in 26 patients with low cardiac index whereas increased in 7 patients with high cardiac index. CI, cardiac index. * *p* < 0.0.5 by Wilcoxon signed-rank test.

**Figure 4 medicina-57-01008-f004:**
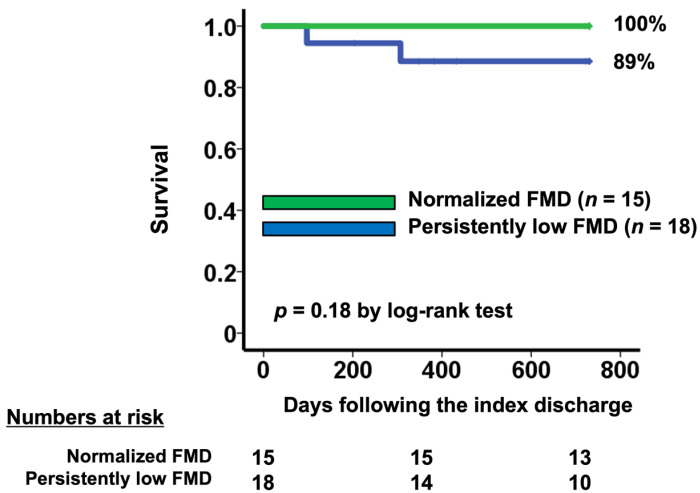
Two-year survival stratified by the change in FMD following TAVI. Patients with persistently low FMD had worse survival compared with those with normalized FMD.

**Table 1 medicina-57-01008-t001:** Baseline characteristics.

	Total (*n* = 51)	Baseline Normal FMD (>6%) (*n* = 18)	Baseline Low FMD (≤6%) (*n* = 33)	*p* Value
Demographics				
Age, years	86 (81, 89)	85 (82, 89)	87 (82, 89)	0.95
Men	12 (24%)	1 (6%)	11 (33%)	0.024 *
Body surface area, m^2^	1.39 (1.34, 1.51)	1.37 (1.30, 1.44)	1.43 (1.36, 1.56)	0.083
Comorbidity				
Diabetes mellitus (%)	8 (16%)	2 (11%)	6 (18%)	0.41
Dyslipidemia (%)	29 (57%)	10 (56%)	19 (58%)	0.56
Current smoker (%)	10 (20%)	1 (6%)	9 (27%)	0.062
Chronic obstructive pulmonary disease (%)	5 (10%)	1 (6%)	4 (12%)	0.42
History of stroke (%)	6 (12%)	2 (11%)	4 (12%)	0.65
Ischemic heart disease (%)	10 (20%)	3 (17%)	7 (21%)	0.5
Peripheral artery disease (%)	17 (33%)	7 (39%)	10 (30%)	0.38
Hemodynamics				
Systolic blood pressure, mmHg	109 (100, 118)	109 (101, 116)	110 (101, 122)	0.82
Heart rate, bpm	72 (63, 79)	72 (61, 75)	73 (64, 81)	0.36
Mean right atrial pressure, mmHg	6 (3, 8)	4 (2, 7)	6 (3, 8)	0.22
Pulmonary capillary wedge pressure, mmHg	12 (8, 16)	12 (7, 17)	13 (9, 16)	0.87
Cardiac index, L/min/m^2^	2.64 (2.28, 2.96)	2.61 (2.38, 2.91)	2.63 (2.25, 2.85)	0.4
Laboratory				
Hemoglobin, g/dL	11.0 (10.0, 12.2)	11.2 (10.2, 12.1)	11.0 (9.2, 11.9)	0.9
Serum albumin, mg/dL	3.9 (3.6, 4.0)	3.9 (3.8, 4.2)	3.8 (3.6, 4.0)	0.37
Serum sodium, mEq/L	140 (139, 142)	140 (140, 141)	141 (139, 142)	0.59
eGFR, mL/min/1.73 m^2^	47.4 (38.0, 64.4)	49.6 (40.8, 64.2)	46.1 (30.8, 60.1)	0.23
Serum C-reactive protein, mg/dL	0.06 (0.03, 0.22)	0.08 (0.04, 0.15)	0.09 (0.03, 0.26)	0.63
Plasma B-type natriuretic peptide, pg/mL	297 (114, 620)	443 (190, 667)	292 (109, 586)	0.32
Echocardiography				
Maximum velocity at aortic valve, m/s	4.7 (4.0, 5.4)	5.1 (4.3, 6.0)	4.5 (4.0, 5.0)	0.038 *
Left ventricular end-diastolic diameter, mm	45 (41, 49)	45 (42, 50)	45 (39, 49)	0.51
Left ventricular ejection fraction, %	65 (57, 72)	65 (53, 68)	64 (59, 72)	0.7
Scoring				
STS score for SAVR, %	5.1 (3.5, 7.9)	4.9 (3.6, 5.9)	5.1 (3.6, 8.4)	0.31
EURO II score, %	3.3 (2.2, 4.3)	3.4 (2.5, 4.1)	3.2 (2.2, 4.6)	0.91
Medication				
Beta-blocker (%)	16 (31%)	5 (28%)	11 (33%)	0.47
Angiotensin converting enzyme II inhibitor (%)	7 (14%)	1 (6%)	6 (18%)	0.21
Renin-angiotensin system inhibitor (%)	37 (73%)	12 (67%)	25 (76%)	0.49
FMD, %	5.3 (3.2, 6.7)	7.5 (6.7, 10.2)	4.5 (2.6, 5.2)	<0.00 *

STS, Society of Thoracic Surgeons; SAVR, surgical aortic valve replacement; FMD, flow-mediated vasodilatation. Data given as mean (25–75 percentile) or number and percentage. Variables were compared between two groups by Wilcoxon signed-rank test as appropriate. * *p* < 0.05.

**Table 2 medicina-57-01008-t002:** Clinical parameters before and after TAVI.

	Baseline (*n* = 51)	After TAVI (*n* = 51)	*p* Value
Systolic BP, mmHg	110 (100, 118)	117 (104, 131)	0.036 *
Diastolic BP, mmHg	58 (49, 65)	56 (50, 62)	0.307
Pulse pressure, mmHg	52 (143, 60)	61 (46, 74)	0.003 *
Pulse Rate, bpm	72 (63, 79)	71 (63, 78)	0.386
sCr, mg/dl	1.00 (0.70, 1.18)	0.95 (0.67, 1.13)	0.274
BNP, pg/ml	368 (114, 620)	138 (41, 178)	<0.001 *
6min. walking distance, m	219 (162, 280)	235 (169, 297)	0.023 *
Echocardiography			
Left ventricular end-diastolic dimension, mm	45 (41, 49)	44 (41, 48)	0.746
Left ventricular ejection fraction, %	65 (57, 72)	67 (59, 70)	0.450
Aortic valve indices			
Peak velocity of aortic valve flow, m/s	4.7 (4.0, 5.4)	2.3 (2.1, 2.6)	<0.001 *
Aortic valve area, cm^2^	0.57 (0.43, 0.69)	1.47 (1.20, 1.70)	<0.001 *
FMD, %	5.3 (3.2, 6.7)	6.3 (4.7, 8.1)	<0.001 *

BP, blood pressure; sCr, serum creatinine; BNP, type B natriuretic peptide; FMD, flow-mediated vasodilatation. Data given as mean (25 percentile–75 percentile). Variables were compared between two groups by Wilcoxon signed-rank test as appropriate. * *p* < 0.05.

**Table 3 medicina-57-01008-t003:** Predictors of the normalization of FMD following TAVI.

	Univariate Analyses		Multivariate Analyses	
	Odds Ratio (95% CI)	*p* Value	Odds Ratio (95% CI)	*p* Value
Age, years	1.01 (0.87–1.16)	0.95		
Men	0.57 (0.13–2.53)	0.46		
Body surface area, m^2^	0.12 (0.01–5.73)	0.28		
Diabetes mellitus	1.86 (0.29–11.9)	0.51		
Dyslipidemia	0.83 (0.21–3.35)	0.8		
Current smoker	4.14 (0.71–24.2)	0.21		
Chronic obstructive pulmonary disease	2.80 (0.26–30.2)	0.40		
History of stroke	2.80 (0.26–30.2)	0.40		
Ischemic heart disease	7.00 (0.74–66.6)	0.21		
Peripheral artery disease	1.38 (0.31–6.20)	0.68		
Systolic blood pressure, mmHg	0.97 (0.92–1.02)	0.27		
Heart rate, bpm	1.01 (0.96–1.08)	0.64		
Mean right atrial pressure, mmHg	1.00 (0.81–1.23)	0.97		
Pulmonary capillary wedge pressure, mmHg	1.07 (0.93–1.24)	0.36		
Cariac index, L/min/m^2^	3.26 (0.56–19.0)	0.19	11.8 (1.12–124)	0.040 *
Hemoglobin, g/dL	0.90 (0.61–1.34)	0.61		
Serum albumin, mg/dL	1.21 (0.19–7.55)	0.84		
Serum sodium, m Eq/L	0.97 (0.85–1.27)	0.85		
eGFR, mL/min/1.73 m^2^	1.00 (0.97–1.05)	0.83		
Serum C-reactive protein, mg/dL	0.50 (0.08–3.26)	0.47		
Plasma B-type natriuretic peptide, pg/mL	1.00 (1.00–1.01)	0.095	1.00 (1.00–1.01)	0.060
Maximum velocity at aortic valve, m/s	2.00 (0.75–5.39)	0.17	1.15 (0.34–3.90)	0.83
Left ventricular end-diastolic diameter, mm	0.95 (0.85–1.06)	0.37		
Left ventricular ejection fraction, %	0.97 (0.90–1.04)	0.40		
STS score	0.98 (0.81–1.18)	0.83		
EURO II score	1.08 (0.86–1.35)	0.53		
Beta blocker	1.00 (0.23–4.28)	1.0		
Angiotensin converting enzyme II inhibitor	5.39 (0.55–52.4)	0.15	7.19 (0.49–106)	0.15

eGFR, estimated glomerular filtration rate; STS, Society of Thoracic Surgeons. Data are presented as odds ratio and 95% confidence interval (CI). * *p* < 0.05.

## Data Availability

Data will be provided by the corresponding author upon reasonable request.

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
