# Peer review of "Improvement in Vascular Endothelial Function following Transcatheter Aortic Valve Implantation"

_medicina, 2021, doi:10.3390/medicina57101008_

Round 1

Reviewer 1 Report

Dear authors,

we congratulate to the nice work: Endothelial dysfunction was measured by Flow-Mediated Dilatation (FMD) in 51 patients after TAVI.  After 1 week follow up FMD improved significantly. Higher cardiac index was associated with normalized FMD. In the multivariate analysis a higher cardiac index at baseline was an independent predictor of FMD normalization following TAVI. Patients with baseline cardiac index (calculated cutoff 2.94 L/min/m2) above the cutoff enjoyed significant improvement in FMD. The authors concluded: Endothelial function, assessed by using FMD, improved immediately after TAVI. Preserved cardiac output might be a key to improve FMD following TAVI.

However, there are some drawbacks in the manuscript that hinder its immediate publication. But in detail:

Language: good. cm2 should be changed to cm2. References in the text must be placed before the dot at the end of the sentence and not after it. This must be changed throughout the text, e.g. “…., and poor outcomes. [1]…” change to “…., and poor outcomes [1].…”

Abstract: The abstract needs to be better structured. The methods and results should be listed separately. In addition, it is not immediately clear from the abstract from which methods the results were found. As an author and reviewer, one has of course gone very deep into the topic. However, the reader should be able to read the abstract in a relaxed manner, feel addressed by it and be able to understand it immediately. More text could possibly help.

Methods: Good structured. Only, 51 TAVIs in 3.5 years is actually a low surgical number, even if the procedure was performed by only one surgeon. Were more patients operated on during that period and only 51 included in the study? Anyway, this needs to be described, e.g. in one or two sentences.

Results: The in-hospital and all-cause mortality needs to be described more clearly. Only the follow up of 33 patients is listed (Fig. 4).

As described in the introduction, there is a low value of FMD associated with smoking. In table 1 (baseline characteristics) there is a clear but not significant difference (p=0.062) between the groups regarding current smoker. Is there a influence to results? What is the distribution of smokers and their influence on the result, if we look at the 33 patients with low FMD in normalized (n=15) and persistently low (n=18)?

Discussion: There is more current literature that has examined the topic as well. The literature should be included and discussed: e.g. Andrea Comella et al. Patients with aortic stenosis exhibit early improved endothelial function following transcatheter aortic valve replacement: The eFAST study, International Journal of Cardiology,Volume 332, 2021, https://doi.org/10.1016/j.ijcard.2021.03.062.

In addition, the article from Moscarelli should be included and discussed: Marco Moscarelli The effect of surgical versus transcatheter aortic valve replacement on endothelial function. An observational study, International Journal of Surgery, 2019, https://doi.org/10.1016/j.ijsu.2019.01.014.

Tables and Figures: The distance of the text and the tables or figures should be larger. In the text, for example, the heading of the tables is closer to the text than to the tables. That should be changed.

Tables: The heading of the table should be more precise, e.g. "Baseline normal FMD" change to "Baseline normal FMD (>6%)" and lower FMD (<6%). Table 2: heading “Baseline (n=51)”

Figures:  The figure should be larger throughout the text.

Literature: Please add doi- numbers.

Reviewer 2 Report

Dear Authors,

Thank you for submitting your paper to the journal. The paper evaluated endothelial disfunction in AS paper after TAVI. It has clinical relevance and it is well presented. It would bring higher relevance if the authors have added the female information as well, and not only showing males data.

Author Response

Response to Reviewer 2 Comments

--Dear Authors,

Thank you for submitting your paper to the journal. The paper evaluated endothelial disfunction in AS paper after TAVI. It has clinical relevance and it is well presented. It would bring higher relevance if the authors have added the female information as well, and not only showing males data.

---Response: We express our great appreciation for the reviewer’s comment. We attempted our best to correct all typographs and grammatical errors to improve our draft. We revised further to respond to the reviewer 1’s comments. In this cohort, most of the participants were female (38/51). Most of the patients in the baseline normal FMD were female (17/18). The progression process of endothelial dysfunction might be different between men and women, and further sub-analysis between men and women might bring us further insights. On the contrary, the impact of sex difference on improvement in FMD following TAVI was not significant in the logistic regression analysis (p = 0.46, see Table 2). 

Before: None

After: Fifth, most of the patients in the baseline normal FMD were female (17/18). The progression process of endothelial dysfunction might be different between men and women, and further sub-analysis between men and women might bring us further insights.

Round 2

Reviewer 1 Report

Dear colleaques,

thank you very much for the revision of the mauscript. However, there are still some small aspects that should be improved:
1. please change the cm2 to cm superscript
2. the visual layout of the article should also be revised, i.e. the spacing of the individual sections should be visually better assigned. 
3. the article should appear in justified text.